# Assessing the real implications for $CO_2$ as generation from renewables increases

Dhruv Suri [1] ✉, Jacques de Chalendar [1] & Inês M. L. Azevedo [1,2,3,4,5,6] ✉

Wind and solar electricity generation account for 14% of total electricity generation in the United States and are expected to continue to grow in the next decade. While increased renewable penetration reduces system-wide emissions, the intermittent nature of these resources disrupts conventional thermal plant operations. Generation displacement exhibits a nonlinear relationship, as thermal units forced to operate at suboptimal levels experience efficiency penalties. Here we show that as renewable generation rises, thermal plants often operate sub-optimally, increasing emissions when forced to respond to variability. Using hourly emissions and generation data from California and Texas, we find that solar and wind energy significantly reduce expected emissions under normal operating conditions - by 92.6% in California and 91.1% in Texas. However, if renewables force plants to operate inefficiently, emissions from natural gas and coal plants could increase by 12% to 26%. These results highlight the complex interactions between renewable energy growth and thermal plant emissions, indicating that careful management of renewables integration is crucial to minimizing overall system-level $CO_2$ emissions, especially in electricity grids with inflexible thermal capacity.

Deep decarbonization of the power sector is critical to limiting the impacts of climate change and advancing sustainable energy transitions. A key component of this effort is the large-scale deployment of renewable energy sources (RES), such as solar and wind, which are rapidly transforming electricity systems worldwide[1]. In response, many regions have adopted ambitious decarbonization targets to accelerate renewable integration. This study focuses on two major U.S. electricity markets: the California Independent System Operator (CAISO) and the Electric Reliability Council of Texas (ERCOT). These markets are notable for both their high annual electricity demand and significant penetration of renewables. California and Texas have also enacted distinct policy frameworks to support decarbonization[2–6]. In California, Senate Bill 100 mandates that 100% of electricity retail sales be supplied by renewable and zero-carbon resources by 2045[7]. Texas, by contrast, was an early adopter of a renewable portfolio standard and has expanded its renewable capacity through a market-based approach[8].

As of 2021, Texas and California were the first- and second-largest U.S. states in terms of electricity consumption[9]. While California's overall demand has declined in part due to efficiency and conservation measures[10], both states have experienced substantial growth in wind and solar generation. As shown in Table 1, renewable energy accounted for 28.4% of annual generation in California and 25.4% in Texas, a significant increase from 2010 levels, when wind and solar represented only 2.9% and 0.4% of California's net generation, respectively[11,12]. Over the same period, Texas increased its share of generation from wind and solar from 6.35% and 0.002–20.7% and 3.1%, respectively.

Earlier studies examining the operational impacts of RES focused primarily on simulations rather than actual historical data. Makarov et al.[13] predicted that CAISO would need additional load-following capacity (0.8 GW up, 0.6 GW down) to integrate 4.1 GW of new wind generation from 2006 to 2010. Katzenstein and Apt[14] modeled a wind or solar photovoltaic plus gas system, finding that steam injection gas

[1]Department of Energy Science & Engineering, Stanford University, Stanford, CA, USA. [2]Precourt Institute for Energy, Doerr School of Sustainability, Stanford University, Stanford, CA, USA. [3]Woods Institute for the Environment, Doerr School of Sustainability, Stanford University, Stanford, CA, USA. [4]Civil and Environmental Engineering, School of Engineering, Stanford University, Stanford, CA, USA. [5]Nova Business School, Carcavelos, Portugal. [6]Department of Earth System Science, Stanford University, Stanford, CA, USA. ✉e-mail: surid@stanford.edu; iazevedo@stanford.edu

**Table 1 | Regional sources of electricity generation, emissions, and emissions intensity for the California Independent System Operator (CAISO) and the Electric Reliability Council of Texas (ERCOT)**

| | California (CAISO) | | | Texas (ERCOT) | | |
|---|---|---|---|---|---|---|
| | **2010** | **2015** | **2021** | **2010** | **2015** | **2021** |
| **Natural gas** | | | | | | |
| Number of plants | 213 | 286 | 298 | 116 | 165 | 216 |
| Installed capacity (GW) | 38.28 | 49.61 | 33.83 | 63.37 | 89.97 | 63.41 |
| Share of generation (%) | 51.32 | 47.94 | 47.39 | 45.23 | 46.97 | 45.28 |
| Share of emissions (%) | 88.79 | 94.98 | 95.42 | 32.83 | 39.41 | 48.28 |
| Average emissions intensity (tons $CO_2$ per MWh) | 0.5 | 0.47 | 0.55 | 0.49 | 0.47 | 0.58 |
| Standard deviation of emissions intensity (tons $CO_2$ per MWh) | 0.63 | 0.33 | 1.27 | 0.18 | 0.20 | 0.20 |
| Average capacity factor (%) | 0.39 | 0.33 | 0.41 | 0.32 | 0.31 | 0.20 |
| Standard deviation of capacity factor (%) | 0.33 | 0.31 | 0.34 | 0.25 | 0.25 | 0.24 |
| **Coal** | | | | | | |
| Number of plants | 7 | 3 | 1 | 16 | 16 | 11 |
| Installed capacity (GW) | 2.01 | 1.74 | 0.06 | 21.09 | 26.33 | 16.53 |
| Share of generation (%) | 1.32 | 0.19 | 0.18 | 35.35 | 26.03 | 18.37 |
| Share of emissions (%) | 4.82 | 0.52 | 0.45 | 66.74 | 59.49 | 50.82 |
| Average emissions intensity (tons $CO_2$ per MWh) | 0.69 | 0.53 | 0.52 | 1.08 | 1.06 | 0.96 |
| Standard deviation of emissions intensity (tons $CO_2$ per MWh) | 0.38 | - | - | 0.27 | 0.09 | 0.33 |
| Average capacity factor (%) | 0.62 | 0.6 | 0.56 | 0.69 | 0.47 | 0.54 |
| Standard deviation of capacity factor (%) | 0.29 | - | - | 0.18 | 0.14 | 0.22 |
| **Utility scale solar** | | | | | | |
| Number of plants | 0 | 483 | 595 | 0 | 39 | 104 |
| Installed capacity (GW) | 0.0 | 14.37 | 15.53 | 0.0 | 2.82 | 8.92 |
| Share of generation (%) | 0.0 | 11.22 | 19.5 | 0.0 | 0.18 | 3.53 |
| Average capacity factor (%) | - | 0.22 | 0.21 | - | 0.17 | 0.18 |
| Standard deviation of capacity factor (%) | - | 0.09 | 0.09 | - | 0.08 | 0.08 |
| **Utility scale wind** | | | | | | |
| Number of plants | 0 | 129 | 108 | 0 | 146 | 169 |
| Installed capacity (GW) | 0.0 | 6.17 | 6.15 | 0.0 | 26.39 | 32.23 |
| Share of generation (%) | 0.0 | 7.92 | 8.91 | 0.0 | 13.75 | 21.90 |
| Average capacity factor (%) | - | 0.25 | 0.27 | - | 0.32 | 0.32 |
| Standard deviation of capacity factor (%) | - | 0.1 | 0.11 | - | 0.10 | 0.11 |

Data are obtained from the United States Environmental Protection Agency's (EPA) Emissions and Generation Resource Integrated Database (eGRID)[41].

generators achieved only 30–50% of expected NOx emissions reductions and close to 80% of expected $CO_2$ emissions reductions.

Others, like Eser et al.[15], show the effect of increased penetration of RES on thermal power plants in Central Western and Eastern Europe. Using a power flow model, the authors show that an increase in penetration of RES induces a 4–23% increase in the number of starts in conventional plants, and a 63–181% increase in load ramp. Eser et al.'s[15] findings are similar to those from other studies regarding the effect of cycling on power plant operational life: not only does increased cycling result in more wear and tear on gas generators[16–18] but also reduces emissions performance, especially at lower capacity factors[19]. Valentino et al. show the change in the number of startups and the degradation of plant heat rate with lower utilization[20]. They use a dispatch model to determine active units and estimate a fuel consumption function based on different blocks of heat rate data. However, the study does not consider heat rate as a function of plant type, ramping, vintage, and degree of cycling, which is a key determinant of heat rates at low utilization levels.

Some studies focus on how RES may change the unit commitment and economic dispatch of generators and conclude that increasing wind and solar penetration changes market dynamics in both nodal and wholesale electricity market designs[21–23]. Other analyses find that RES lead to an increase in ramping, lower utilization, and higher

operations and maintenance costs[24–28]. While unit commitment and economic dispatch studies using simulations provide valuable insights and demonstrate that increasing penetration of RES changes several operational characteristics of thermal plants, these approaches may miss real-world operations in the electricity system, including coping with transmission constraints, with policies, and with RES variability[29]. Empirical approaches that rely on observational data mitigate some of these limitations, but may not capture future conditions with larger RES capacity and generation. Examples of such approaches include the studies by Fell and Johnson[29], Mills et al.[30], and Wiser et al.[31]. These studies find that renewables do indeed displace thermal generation; however, the nature and magnitude of the displacement are still heterogeneous across the studies. Empirical studies examining how renewable energy affects emissions and ramping in thermal power plants remain limited. In particular, there is little analysis of how operational changes drive emissions intensity and how these effects can be mitigated through improved dispatch.

In this study, we develop a framework to evaluate the operational and environmental impacts of renewable integration on thermal generation. Using historical data from 2018 to 2023 for CAISO and ERCOT, we analyze patterns in thermal plant behavior under increasing renewable penetration. We begin by comparing emissions outcomes under simplified operational scenarios. We then use regression

analysis to estimate the system-level change in generation and emissions associated with a 1% increase in wind and solar generation. Finally, we examine individual plant-level responses, highlighting heterogeneous effects by region and fuel type.

## Results

### Alternate emissions scenarios for thermal power plants

We use hourly thermal generation and $CO_2$ emissions from the United States Environmental Protection Agency's (US EPA) Continuous Emissions Monitoring System (CEMS) at the generating unit and power plant level to study the operational characteristics of power plants. We start by assessing the observed emissions in each year from 2018 to 2023 in CAISO and ERCOT, and then compare those emissions to what they would have been under different operational scenarios. Specifically, we compare the historical emissions to counterfactual scenarios in which each power plant operates at its observed emissions intensity under both low and high capacity factor conditions.

The rationale for constructing simplified emissions scenarios is illustrated in Fig. 1, which presents the relationship between emissions intensity and capacity factor across generating units at two representative thermal power plants: Moss Landing, a natural gas-fired facility in CAISO (Fig. 1a), and Tolk Station, a coal-fired plant in ERCOT (Fig. 1b). In both cases, operation at lower capacity factors is associated with a marked increase in emissions intensity. For example, reducing the capacity factor from 0.3 to 0.05 results in a 133% increase in average emissions intensity at Moss Landing and a 51% increase at Tolk Station. This pattern reflects a broader trend across thermal generating units: emissions intensity rises as capacity factor declines. The underlying mechanism is thermodynamic inefficiency—when thermal units operate at low utilization levels, they consume more fuel per unit of electricity generated, leading to higher $CO_2$ emissions per megawatt-hour. Each point in the figure represents an hourly observation of generation and emissions at the unit level, as reported to CEMS.

At Moss Landing, generating units operating at capacity factors greater than 0.3 exhibit a mean emissions intensity of 0.36 tons $CO_2$ per MWh. When operating below a capacity factor of 0.05, this value increases to 0.84 tons $CO_2$ per MWh. At Tolk Station, the mean emissions intensity increases from 0.94 to 1.43 tons $CO_2$ per MWh over the same capacity factor range. These findings are not limited to the two facilities examined. Across the full fleet of natural gas and coal-fired units in CAISO and ERCOT, we observe consistent increases in emissions intensity at lower capacity factors. Supplementary Figs. 3

through 11 present analogous plots for all relevant generating units. To account for this systematic variation in operating emissions, we construct alternative emissions scenarios as described below.

As the generation from RES increases, more thermal plants will be required to operate at low capacity factors or will remain online as spinning reserves given ramping and startup constraints (also shown in Supplementary Fig. 12). Low utilization will in turn result in higher hourly heat rates as a direct consequence of which system-level emissions will also increase. To model these effects, we start by considering two simplified emissions intensity scenarios that represent different operational regimes of thermal plants: (i) a "low emissions" scenario where we select, for each plant, the 10th percentile of emissions intensity, (ii) and a "high emissions" scenario where we consider the emissions intensity of the plant at the 90th percentile of their observed emissions intensity in CEMS. These low and high emissions intensity scenarios are related to how the plants are operated, and thus, in the next results sections, we will address this issue using regression-based models. Figure 2 presents annual emissions under three scenarios: low, high, and observed emissions intensity. Figure 2b, d and f additionally show the distribution of plant-level emissions intensities, highlighting the variability among units with the same fuel type within each Independent System Operator (ISO) region.

These simplified scenarios provide important bounding results: if power plants were to be operating far from their optimal operation limits, i.e., with low capacity factor and high emissions intensity, the overall $CO_2$ emissions from natural gas and coal power plants in CAISO and ERCOT would be 12 and 26% higher than the observed emissions (depending on the year). Although the emissions intensity at low capacity factor operation is several factors higher than the baseline emissions intensity, plants are operated infrequently at very low capacity factors (<0.05). Hence, the 90th percentile of emissions intensity is lower than the extreme values observed in Fig. 1.

### Marginal impacts of wind and solar on plant-level generation, emissions and emissions intensity

Now, we turn our attention to providing more systematic estimates on the effect of renewables on thermal power plant operations, using entity and time fixed-effects panel regression model with a logarithmic specification. This model describes the change in the dependent variable (generation, emissions, or emissions intensity) as a function of the generation from wind and solar, residual thermal demand, individual power-plant entity effects, time effects, and other effects from control variables. The logarithmic specification used is given by Eq. (1).

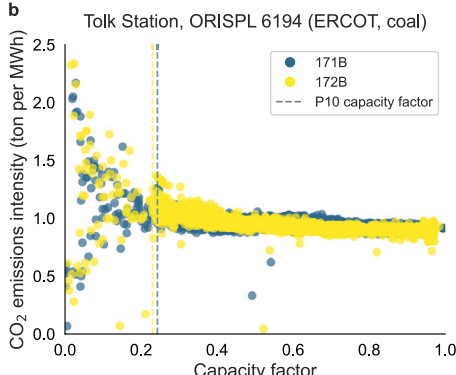

**Fig. 1 | Nonlinear relationship between emissions intensity and capacity factor for thermal power plants. a** Moss Landing, a natural gas-fired power plant in the California Independent System Operator (CAISO) market. **b** Tolk Station, a coal-fired power plant in the Electric Reliability Council of Texas (ERCOT) market. Hourly emissions intensity (tons of $CO_2$ per MWh) is plotted against unit-level capacity factor for each facility. Colored points represent individual generating

units, identified using unique plant and unit codes from the Office of Regulatory Information Systems Plant Location (ORISPL) database maintained by the U.S. Energy Information Administration. Dashed lines denote the 10th percentile (P10) of the capacity factor distribution for each unit, indicating low-utilization thresholds. The results highlight the nonlinear increase in emissions intensity under partial-load operating conditions in thermal generation assets.

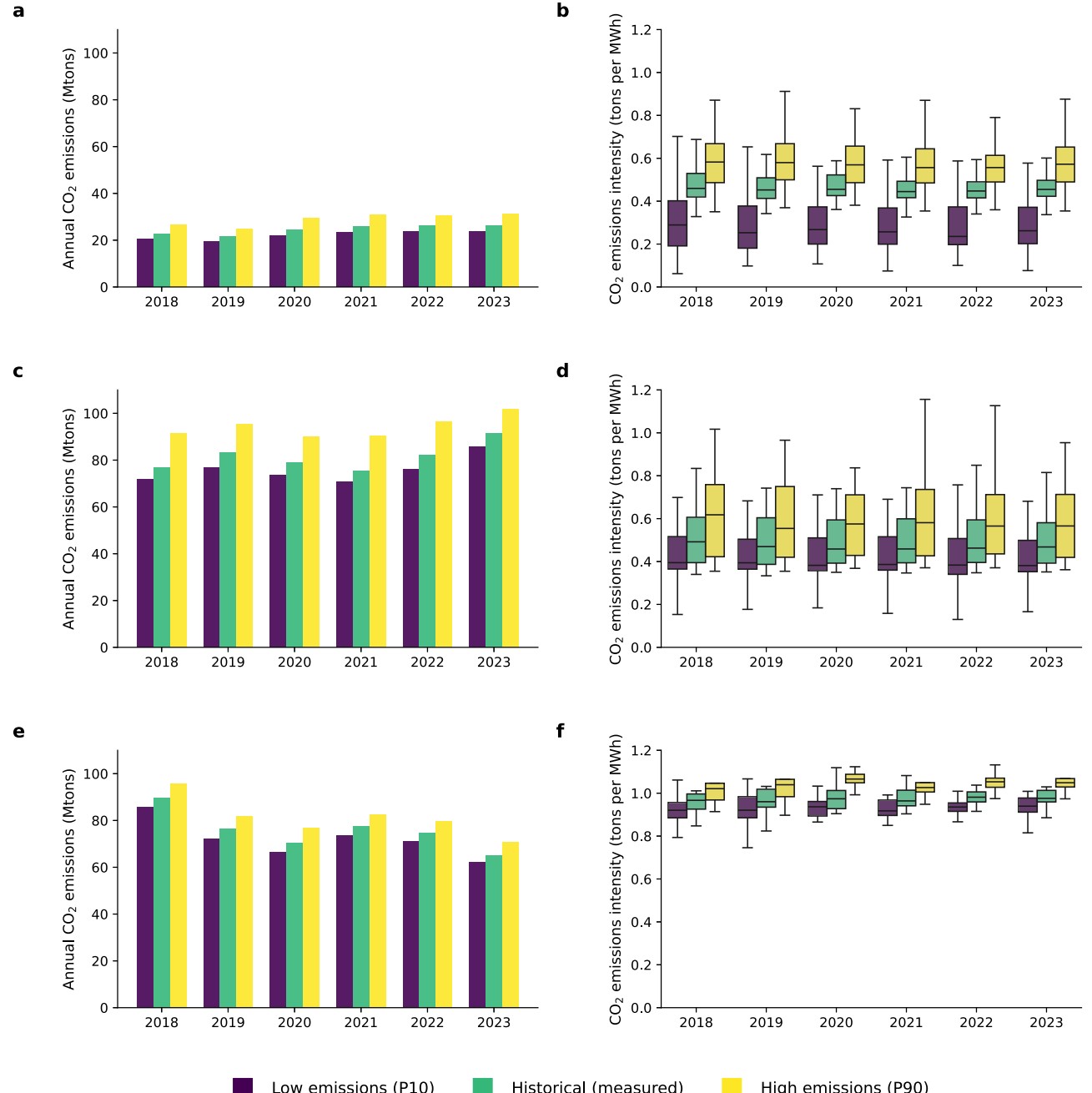

**Fig. 2 | Alternate CO$_2$ emissions scenarios for thermal plants.** Three scenarios representing different operational regimes of thermal plants for **a**, **b** Natural gas plants in CAISO, **c**, **d** Natural gas plants in ERCOT and **e**, **f** Coal plants in ERCOT. **a**, **c**, and **e** show annual emissions under three scenarios: (i) A "low emissions" scenario where plants operate at the 10th percentile of their emissions intensity, (ii) historical measured emissions, and (iii) a "high emissions" scenario where plants operate at the 90th percentile of their observed emissions intensity. **b**, **d**, and **f** show boxplots describing the range of individual power plant emissions intensities under the three scenarios, where the box shows the interquartile range (25th–75th percentile), the line within the box indicates the median, and the whiskers extend to the minimum and maximum values, excluding outliers.

The control variables include the total wind and solar generation and electricity demand from the ISO's trading partners, as well as a measure of wind generation intermittency. Wind intermittency is defined as the intra-day variability in output and is calculated by taking the absolute value of the differences in hourly wind generation and summing them over the day. This captures short-term fluctuations in wind availability, which may influence the dispatch of thermal resources. Rather than aggregating renewable generation, we model wind and solar as separate independent variables to reflect their distinct temporal profiles and differing penetration levels in CAISO and ERCOT.

Time fixed effects are included at the month and year level to account for temporal trends, while plant fixed effects control for time-invariant characteristics of each generating unit. These controls allow us to isolate the relationship between renewable generation and thermal plant operation across both systems. Under the logarithmic specification, the estimated coefficients can be interpreted as the percent change in the dependent variable—generation, emissions, or emissions intensity—associated with a 1% change in wind or solar generation.

In Table 2, we show the coefficients of the panel regression model for CAISO and ERCOT. In this way, we are able to quantify the percent

**Table 2 | Coefficients from the panel regression specification with generation, CO$_2$ emissions, and CO$_2$ emissions intensity as the dependent variables**

| | CAISO | | | ERCOT | | |
|---|---|---|---|---|---|---|
| | Generation | Emissions | Emissions intensity | Generation | Emissions | Emissions intensity |
| **Net demand** | 2.01*** | 2.91*** | −0.10*** | 1.88*** | 1.78*** | −0.10*** |
| | (0.044) | (0.04) | (0.006) | (0.042) | (0.039) | (0.006) |
| **Solar** | −0.27*** | −0.25*** | 0.02*** | −0.03*** | −0.03*** | 0.0004*** |
| | (0.031) | (0.028) | (0.004) | (0.008) | (0.008) | (0.001) |
| **Wind** | −0.23*** | −0.22*** | 0.01*** | −0.34*** | −0.31*** | 0.02*** |
| | (0.012) | (0.004) | (0.002) | (0.007) | (0.007) | (0.001) |
| **Wind ramp** | 0.12*** | 0.12*** | −0.003 | 0.12*** | 0.11*** | −0.01*** |
| | (0.119) | (0.013) | (0.002) | (0.009) | (0.009) | (0.001) |
| **Solar (ext)** | 0.01 | 0.01 | −0.003 | 0.01 | 0.01 | −0.0003 |
| | (0.012) | (0.028) | (0.004) | (0.012) | (0.007) | (0.001) |
| **Wind (ext)** | −0.01* | −0.01* | 0.002* | −0.02** | −0.02*** | 0.002 |
| | (0.005) | (0.004) | (0.001) | (0.008) | (0.007) | (0.001) |
| **R-squared** | 0.84 | 0.84 | 0.77 | 0.78 | 0.80 | 0.92 |
| **Num. obs.** | 44,724 | 44,724 | 44,724 | 87,089 | 87,089 | 87,089 |

Each column reports results where the endogenous variable is generation, emissions, or emissions intensity, respectively. Control variables include the sum of wind generation, solar generation, and electricity demand in neighboring balancing areas of the system operator—specifically, the CAISO and the ERCOT—as well as the average daily intermittency of wind generation. Time fixed effects control for the month and year of each daily observation. Significance levels: ***$p < 0.001$, **$p < 0.01$, *$p < 0.05$. Plant fixed-effects control for unobserved heterogeneity across generating units.

change in emissions intensity, total emissions, and generation for thermal power plants attributable to a marginal increase in generation from solar and wind. Here, the term "marginal" implies an additional unit (or unit percent) increase in generation from renewables and its resulting impact. This, in turn, can be used to determine the expected versus actual emissions displacement from renewables.

From Table 2, an increase in wind and solar generation is associated with lower thermal generation and emissions. That said, if this displacement was 100% efficient (or 1:1), both generation and emissions would reduce by an equivalent amount, and thus should have identical coefficients in the fixed-effects model. In CAISO, the displacement from wind and solar is unequal. Our regression estimates indicate that a 1% increase in solar generation is associated with a 0.27% decrease in thermal generation and a 0.25% decrease in total emissions. By comparing the coefficients for solar generation with emissions and emissions intensity as the exogenous variables, we can compute the share of expected emissions displacement due to the logarithmic specification of the panel regression (see "Methods"). Based on these coefficients, the observed emissions reduction associated with solar generation represents 92.6% of the theoretically expected emissions displacement. Similarly, the observed reduction associated with wind generation represents 95.6% of the theoretically expected emissions displacement. In ERCOT, on the other hand, displacement from wind is an order of magnitude more pronounced than the displacement from solar, which may be attributable to the difference in the installed utility-scale capacity of the two resources in the ISO. Examining the relationship between wind generation and displacement, our analysis shows that a 1% increase in wind generation is associated with a 0.34% decrease in thermal generation and a 0.31% decrease in total emissions. Based on these coefficients, the observed emissions reduction associated with wind generation in ERCOT represents 91.1% of the theoretically expected emissions displacement.

While the expected vs actual emissions displacement helps us understand the system-level impact of renewables, it still does not associate the effect of renewables on plant-level emissions intensity. Looking closely at the coefficients of the model with emissions intensity as the dependent variable, we find that a 1% increased generation from wind and solar does indeed impact plant-level emissions intensity for existing thermal generators. For CAISO, thermal power plant intensities are more susceptible to increased generation from solar as compared to wind (0.02% for solar, 0.01% for wind). In ERCOT, generation from wind results in a 0.02% increase in the emissions intensity of plants. While these coefficients may initially appear small, the hourly ramp in solar generation in CAISO during the day can reach 50%, which corresponds to an estimated 1% increase in the emissions intensity of power plants for that hour, according to our model. If we consider CAISO's 2022 system-level emissions averaged uniformly across all hours, a 1% increase in intensity across all power plants is equivalent to 53 tons of additional CO$_2$. As the ramp becomes more pronounced with increasing intermittent renewable capacity, this marginal increase should not be ignored. Meeting 2030 and 2045 clean energy targets in both CAISO and ERCOT will require the installation of more installed renewable capacity, resulting in an even higher percent change in generation from renewables across all days of the year. Supplementary Figure 14 demonstrates the change in CO2 emissions intensity for a 0-100% increase in generation from solar and wind in CAISO and ERCOT.

Intermittency in generation from wind is measured by the daily wind ramp variable, represented by $\overline{W}_t$ in Eq. (1). In contrast to the coefficients for wind and solar generation, higher wind variability is associated with higher generation from thermal plants in both regions. The magnitude of the positive displacement from wind ramping relative to wind generation is also quite significant—52% for CAISO, and 35% for ERCOT. Clearly, wind and solar generation play a key role in determining the generation and emissions in thermal plants, however, variability and intermittency in the generation from renewables has a significant impact as well.

## Individual plant response in CAISO and ERCOT is heterogeneous by fuel type and nameplate capacity

The panel regression formulation helps us understand how thermal plants across the fleet respond to changes in generation from wind and solar. Plant fixed-effects further control for heterogeneity across plant types and fuel inputs, however, this formulation does not yield any insights into how heterogeneous these plants are, and what the impact of increased generation from wind and solar results in on a per-plant basis. To further understand plant-specific response, we develop a fixed-effects ordinary least squares (OLS) regression formulation with the same specification as the panel regression above, except for every

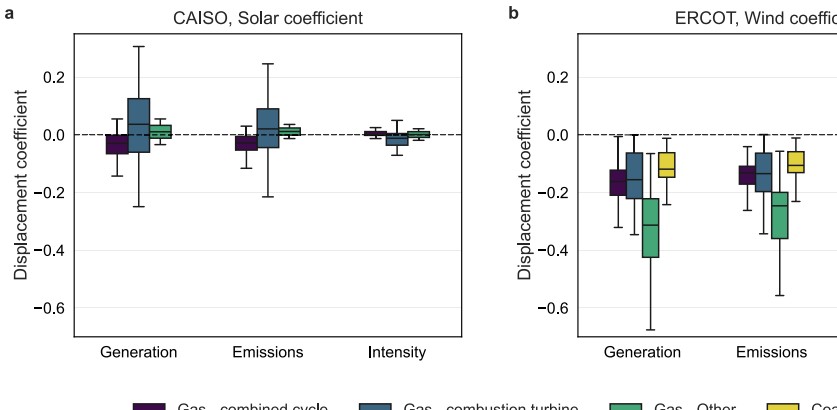

**Fig. 3 | Distribution of displacement coefficients from ordinary least squares (OLS) regressions. a** Estimated coefficients for individual natural gas power plants in the California Independent System Operator (CAISO) market in response to solar generation. **b** Estimated coefficients for coal and natural gas power plants in the Electric Reliability Council of Texas (ERCOT) market in response to wind generation. Plants are categorized by technology type: Gas−combined cycle, Gas−

combustion turbine, Gas−Other, and Coal. Each boxplot summarizes the distribution of plant-level coefficients, with boxes indicating the interquartile range (25th–75th percentile), central lines showing the median, and whiskers extending to the minimum and maximum values excluding outliers. Negative coefficients reflect the displacement of conventional thermal generation by variable renewable energy sources.

individual plant (no panel is considered) and using hourly instead of daily data points. The ramp variable in the formulation is determined as a difference in wind generation at the intra-hourly level. While the panel regression provided a single coefficient for a dependent variable in response to solar and wind, here, we consider a distribution of coefficients across all plants to model their response.

Figure 3 shows the distribution of coefficients for plant generation, emissions, and emissions intensity in response to solar in CAISO (Fig. 3a) and wind in ERCOT (Fig. 3b). A positive coefficient indicates a positive correlation between the dependent variable and renewable generation, and vice versa. We find that natural gas plants in CAISO show highly heterogeneous relationships with solar generation, and the associated change in generation, emissions, and emissions intensity. In contrast, all but two plants in ERCOT demonstrate lower generation and emissions with higher wind generation. However, the emissions intensity of all coal plants and most gas plants shows a positive correlation with marginal increases in wind generation.

The heterogeneous relationship between natural gas plants in CAISO compared to ERCOT may be related to shifting generation patterns accompanying increased utility-scale solar capacity. Fast-ramping peaker plants replace base load combined cycle plants during peak production hours. In CAISO, four out of seven plants with a coefficient greater than 1.0 account for 28% of generation and also exhibit the highest emissions intensity of all plants in the balancing region. In ERCOT, plant behavior is very different from that of CAISO. The mean coefficient for generation is −0.41, which is close to our fleet-level coefficient of −0.34 from the panel regression. Coal plants with a relatively smaller nameplate capacity are characterized by a higher emissions intensity, even though the capacity factor of these plants is comparable to that of larger plants. The two smallest coal plants in ERCOT show the largest deviation in response to wind (−2.25%, −2.07%) and also have a NOx emissions intensity 60% higher than the mean of all coal plants indicating more frequent ramping and startup.

Figure 4 shows the regression coefficients for plant generation, further characterized by nameplate capacity and capacity factor. Each data point represents a power plant. The x axis is annual capacity factor, and the y axis is the OLS coefficient in response to solar generation in CAISO and wind generation in ERCOT. The size of the data point indicates the nameplate capacity of the plant and the color indicates the annual emissions intensity. All plants are natural gas-fired in CAISO, and a mix of natural gas and coal in ERCOT. In CAISO, larger plants exhibit negative correlations with solar generation, although the

magnitude of this correlation is smaller compared to plants with lower nameplate capacity. Smaller plants are also characterized by a significantly lower annual capacity factor, high emissions intensity, and a wider distribution in coefficient magnitude. In ERCOT, the markers in Fig. 4 show a mix of coal and natural gas-fired plants. All coal plants are characterized by a higher emissions intensity and operate at an annual capacity factor consistently greater than 0.3. The coefficients associated with wind generation show less variation for larger plants, with values not falling below −0.2. Similarly to CAISO, natural gas plants with a lower nameplate capacity show the highest emissions intensity and operate at lower annual capacity factors.

Supplementary Fig. 13 shows a comparison between the distrubution of coefficients obtained from the plant-level OLS with emissions displacement estimates from Katzenstein and Apt[14] and Graf et al.[32].

## Discussion

In this study, we analyze the impact of accelerated grid integration of utility-scale wind and solar on the operational characteristics of coal and natural gas plants in two U.S. regions—CAISO and ERCOT. Using hourly power plant generation and emissions data from the CEMS, we find that operating power plants at low capacity factors significantly affects their heat rate and emissions intensity. This observation aligns with prior studies, such as Katzenstein and Apt[14], which demonstrated that increased cycling of fossil fuel plants to accommodate RES leads to higher emissions intensity due to reduced operational efficiency. Contrastingly, our results diverge from those of Fell and Johnson[29], who reported that renewable energy integration resulted in a proportional decrease in emissions without markedly affecting the efficiency of thermal plants. This discrepancy may stem from regional variations in grid management practices, the inherent flexibility of thermal power plants, and the specific methodologies employed in capacity expansion modeling.

At a systems level, we observe a negative correlation between renewable generation and both thermal generation and emissions, though the magnitude of this relationship varies by resource type. In contrast, the emissions intensity of thermal plants increases, albeit marginally, compared to the reduction in overall generation and emissions. If emissions displacement due to renewables were perfectly efficient, the coefficients in our panel regression for generation and emissions would be identical. For instance, if thermal generation at a plant decreases by 0.2% in response to solar, with no impact on the

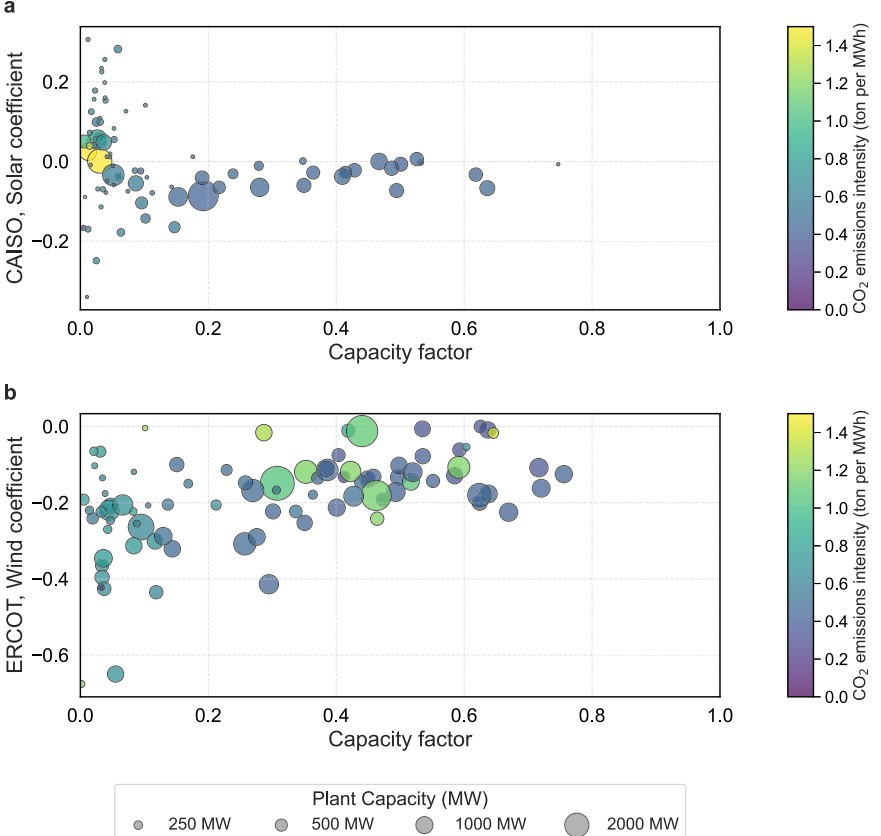

**Fig. 4 | Displacement coefficients as a function of plant-level capacity factor and emissions intensity in 2023. a** Ordinary least squares (OLS) regression coefficients for natural gas-fired power plants in the California Independent System Operator (CAISO) market in response to solar generation. **b** OLS regression coefficients for coal and natural gas-fired power plants in the Electric Reliability Council of Texas (ERCOT) market in response to wind generation. Each marker represents a single power plant. The horizontal axis denotes the annual capacity factor, and the vertical axis shows the estimated displacement coefficient. Marker size corresponds to plant nameplate capacity (in megawatts, MW), and color indicates annual emissions intensity (in tons of $CO_2$ per MWh). In CAISO, smaller and lower-utilization plants tend to have higher emissions intensity and more variable displacement responses. In ERCOT, coal plants operate at higher capacity factors and emissions intensities, while natural gas plants show greater variability in operating characteristics and wind displacement.

heat rate or emissions intensity, emissions should decline by the same proportion. However, we find that the emissions coefficient consistently lags behind the generation coefficient for both wind and solar, indicating that renewables do not displace emissions on a one-to-one basis. Using a logarithmic specification for emissions and emissions intensity, we quantify the extent to which wind and solar reduce expected emissions from thermal plants.

Many capacity expansion models assume uniform heat rates and linear displacement effects for thermal power plants, irrespective of their operational levels. This simplification can introduce inaccuracies, particularly in high-renewable scenarios where thermal plants frequently operate at suboptimal capacity factors. Models such as OSeMOSYS[33] and GenX[34], while robust in various aspects, often apply constant efficiency parameters to thermal units, overlooking the impact of ramping and part-load operation. Our empirical findings suggest that such assumptions may lead to an underestimation of emissions, as they fail to account for efficiency losses associated with increased cycling and reduced capacity factors.

The observed inefficiencies in thermal plant operations under high renewable penetration have significant policy implications. Policymakers should consider strategies to mitigate these effects, such as investing in energy storage technologies to stabilize plant operations, enhancing grid infrastructure to improve flexibility, and implementing demand-side management programs to align consumption with renewable generation patterns. While it is outside the scope of this

paper to model the implications of different strategies, some strategies may warrant attention in future analysis. For example, storage technologies could be used as a means to sustain constant capacity factors and avoid ramping, and could regulate and even substitute the use of spinning reserves. Similarly, there may be emissions reduction benefits in ramping a few bigger units rather than cycling more frequently a large number of small peaking units. Other strategies may include the expansion of new transmission infrastructure or demand-side management strategies such as demand response and targeted building-level energy efficiency retrofits. For supply-side infra-structure, thermal power plants can be designed to operate efficiently at low capacity factors or part-load operations by improving the efficiency of boilers and steam turbines with variable steam flow rate, or by incorporating natural gas-fired high-temperature fuel cells. Addressing these secondary impacts is critical for consistently lowering emissions in a low-carbon future.

## Methods
We perform our analysis for the years 2018 through 2023 for the CAISO and ERCOT. We start by describing the data used, followed by the methods.

### Data
We develop a dataset with thermal power plants in CAISO and ERCOT that includes hourly unit-level emissions, gross generation, heat rate,

and fuel consumption from the EPA's Air Markets Program Data[35]. The EPA requires all combustion power plants with a nameplate capacity greater than 25 MW to install and maintain a CEMS that records key unit-level operational parameters[36].

We obtain plant nameplate capacity, total annual generation, primary fuel information, and total annual emissions from the EPA's Emissions and Generation Resource Integrated Database (eGRID)[12]. eGRID is a comprehensive source of data on the environmental characteristics of electric power plants and their associated unit stacks. Each data file contains key indicators at different levels of aggregation −unit, generator, plant, state, balancing area, NERC region and eGRID subregion. A manual inspection of the plant-level nameplate capacity indicates that the capacity indicated is not accurate in all cases, whereas the same is not true for aggregated generator-level information. Hence, instead of obtaining the plant nameplate capacity directly, we compute the sum of the generator nameplate capacity (in MW) of all generators associated with a particular plant, as identified by their unique Office of Regulatory Information Systems Plant Code. Hourly data from CEMS is collated with plant nameplate capacity to determine hourly capacity factor values and the associated fuel stack.

Hourly generation from wind, solar, hydro, and geothermal energy is obtained from US Energy Information Administration Form 930 (EIA-930) data. We use processed publicly available data obtained from a physics-informed data reconciliation framework for real-time electricity and emissions tracking developed by Chalendar and Benson[28,37]. Using this dataset, rather than raw EIA-930 data, has two key advantages. First, the reconciled data sets are devoid of unrealistic generation values and contain estimates within reasonable confidence intervals for time intervals where data is missing or erroneous. Second, this method includes an optimization-based data reconciliation to impose physical relations that guarantee energy conservation between key variables. Thus, this dataset provides electric system operating data on generation, consumption, and exchanges of electricity for every hour and at the level of the balancing area.

## Model specification

We use a fixed-effects panel regression formulation to quantify the marginal effect of wind and solar generation on the generation, emissions, and emissions intensity of thermal plants. Similar to Bushnell and Wolfram[38], Graf et al.[32] and Fell and Johnson[29], we use an entity and time fixed-effects logarithmic specification for all plants within an ISO using daily observations in Eq. (1).

$$\ln y_{i,t} = \alpha_i + \beta_1 \ln D'_t + \beta_2 \ln S_t + \beta_3 \ln W_t \\ + \beta_4 \ln \overline{W}_t + \gamma \ln \mathbf{X}_t + \eta_{t,m} + \eta_{t,y} + \alpha_i \times \eta_{t,m} + \epsilon_i \tag{1}$$

where $y_{i,t}$ is the dependent variable of interest, which is daily generation in MWh, emissions in tonnes or emissions intensity in tonnes/ MWh for power plant $i$ at timestep $t$. $D_t$, $S_t$ and $W_t$ represent the daily net thermal generation, generation from solar, and generation from wind in MWh for the ISO to which $i$ belongs. The daily net thermal generation is the total ISO thermal generation minus generation from hydro and imports, i.e., $D'_t = D_t - H_t - I_t$, where $D_t$, $H_t$ and $I_t$ represent the daily thermal generation, generation from hydro and net imports in MWh for the ISO. $\overline{W}_t$ is a parameter that estimates the intermittency of net generation from wind at the hourly level and is calculated by taking the sum of the absolute value of hourly differences in wind output over the course of the day. Mathematically, this is expressed as $\overline{W}_t = \sum_{h=0}^{23} |W_{t,h+1} - W_{t,h}|$. The intermittency parameter $\overline{W}_t$ captures the cumulative hourly fluctuations in wind generation throughout the day, rather than using a simpler measure like daily variance. This specification focuses on short-term variations between consecutive hours that may require rapid adjustments in thermal generation to maintain grid stability. While variance would measure overall daily deviations from mean wind generation, our intermittency measure

specifically quantifies the magnitude of hour-to-hour changes that operators must balance. This is particularly relevant as frequent short-term fluctuations in wind generation may necessitate different operational responses from thermal plants compared to more gradual changes over the course of a day.

$\mathbf{X}_t$ is a set of control variables that includes the sum of wind generation, solar generation, and demand for trading partners of the ISO to which $i$ belongs. For instance, CAISO trades with the other balancing authorities in the Northwest and Southwest NERC subregions. For every hour, these external control variables would consider the sum of wind generation, solar generation, and demand for these two regions. For ERCOT, the only trading partner is the Southwest Power Pool. $\eta_{t,m}$ and $\eta_{t,y}$ are the time fixed-effects parameters controlling for the month and year, respectively. We interact the monthly fixed-effects variable with the indicator variable for the plant ID to account for heterogeneity in fuel prices across different technology categories, regions and plant types. $\epsilon$ represents the error term.

The model specifications employed by Graf et al.[32] and Fell and Johnson[29] may not be able to fully explain the deviation caused due to intermittency in renewable generation due to gaps in granularity and panel formulation. Graf et al.[32] do not include granular regional generation from trading partners, and also conduct the analysis on annual data and hence miss intra-day operational parameters such as resource ramping. Fell and Johnson[29] do not construct a plant-level panel regression but rather run a time fixed-effects model for the balancing authority that does not account for location-based heterogeneity.

In our formulation, the parameters that are of interest are the coefficients for solar, $\beta_2$ and wind, $\beta_3$. These coefficients represent the percent change in the dependent variable per 1% increase in generation from solar and wind, respectively. For example, if the dependent variable $y_{i,t}$ represents the net generation from thermal plants for an ISO, then $\beta_2$ and $\beta_3$ will represent the percent change in generation for thermal plants per unit percent increase in generation from solar and wind. These coefficients thus measure the associated change in the dependent variable under the assumption that daily wind and solar production (represented by $W$ and $S$) are uncorrelated with the error term $\epsilon$ after controlling for net demand, entity, and time fixed effects. As suggested by Qiu et al.[39], this assumption is valid for wind at the hourly level and thus can be extrapolated to daily observations. Solar output is likely to be correlated with the error term and time fixed effects parameter when considering hourly data, which is why we aggregate hourly observations to daily intervals in the statistical model above. To test for correlation between thermal power plant generation and other exogenous variables, in Supplementary Figs. 1 and 2, we present the Pearson's correlation coefficient, along with the Durbin−Watson test to check for autocorrelation in the panel regression model for each entity. We also present alternate specifications of the statistical model in Supplementary Note 1.

## Displacement effectiveness

Modeling the response of individual plants outside a panel regression formulation helps us understand whether plants are individually load-following, solar-following, and wind-following, and the extent of their response. In contrast to the fixed effects ISO-level formulation considering daily time steps, the plant-level model uses hourly data and is given by Eq. (2).

$$\ln y_t = \alpha + \beta_1 \ln D'_t + \beta_2 \ln S_t + \beta_3 \ln W_t + \beta_4 \ln \overline{W}_t + \gamma \ln \mathbf{X}_t + \eta_{t,m} + \eta_{t,y} + \epsilon \tag{2}$$

where the variables are the same as those in the panel regression formulation above. In the plant-level OLS, $\overline{W}_t$ measures the deviation in wind generation from the previous hour.

To determine the share of emissions displaced due to renewables, we use the panel regression formulations where emissions and

emissions intensity are the dependent variables, similar to Graf et al.[32]

$$\ln CO_2EI_{p,t} = \alpha_i + \beta_1 \ln D'_t + \beta_2 \ln S_t + \beta_3 \ln W_t \\ + \beta_4 \ln \overline{W}_t + \gamma \ln \mathbf{X}_t + \eta_{t,m} + \eta_{t,y} + \alpha_i \times \eta_{t,m} + \epsilon_i \tag{3}$$

$$\ln CO_{2,p,t} = \alpha'_i + \beta'_1 \ln D'_t + \beta'_2 \ln S_t + \beta'_3 \ln W_t \\ + \beta'_4 \ln \overline{W}_t + \gamma' \ln \mathbf{X}_t + \eta'_{t,m} + \eta'_{t,y} + \alpha'_i \times \eta'_{t,m} + \epsilon'_i \tag{4}$$

In Eq. (3), the marginal effect due to increased generation from solar can be represented by $\alpha_2$.

$$\alpha_2 = \frac{\partial(\ln CO_2EI_{p,t})}{\partial(\ln solar_t)} \tag{5}$$

$$\alpha_2 = \frac{\partial\left(\ln \frac{CO_{2,p,t}}{G_{p,t}}\right)}{\partial(\ln solar_t)} \tag{6}$$

$$\alpha_2 = \frac{\partial(\ln CO_{2,p,t} - \ln G_{p,t})}{\partial(\ln solar_t)} \tag{7}$$

$$\alpha_2 = \frac{\partial(\ln CO_{2,p,t})}{\partial(\ln solar_t)} - \frac{\partial(\ln G_{p,t})}{\partial(\ln solar_t)} \tag{8}$$

$$\alpha_2 = \alpha'_2 - \frac{\partial(\ln G_{p,t})}{\partial(\ln solar_t)} \tag{9}$$

Rearranging, gives:

$$\frac{\partial(\ln G_{p,t})}{\partial(\ln solar_t)} = \alpha_2 - \alpha'_2 \tag{10}$$

Equation (10) represents the change in thermal generation if generation from solar replaces the generation from fossil-fueled plants on a 1:1 basis. Thus, the fraction of expected emissions reductions for solar is represented by:

$$\text{Fraction of expected emissions reduction (solar)} = \frac{\alpha'_2}{\alpha'_2 - \alpha_2} \tag{11}$$

Similarly, for wind, the expected emissions reductions and the corresponding fraction are given by:

$$\frac{\partial(\ln G_{p,t})}{\partial(\ln wind_t)} = \alpha_3 - \alpha'_3 \tag{12}$$

$$\text{Fraction of expected emissions reduction (wind)} = \frac{\alpha'_3}{\alpha'_3 - \alpha_3} \tag{13}$$

## Data availability
The source and processed data are available on Zenodo[40].

## Code availability
The code to replicate the results is available on Zenodo[40].

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

## Acknowledgements
The authors acknowledge Kirat Singh, Nils Angliviel de la Beaumelle and Kamran Tehranchi for their assistance with reviewing the manuscript. We would like to acknowledge the funding from the Precourt Institute for Energy, Doerr School of Sustainability, Stanford University and from the Energy Science and Engineering Department at Stanford University.

## Author contributions
D.S., J.D.C., and I.M.L.A. conceived and designed the methods. D.S. developed the code and conducted the analysis. D.S. wrote the manuscript with input from J.D.C. and I.M.L.A. J.D.C. and I.M.L.A. examined and edited the manuscript. J.D.C. and I.M.L.A. supervised the research. D.S., J.D.C., and I.M.L.A. contributed to the interpretation and drafting of the paper.

## Competing interests
The authors declare no competing interests.
