## [Transparent Peer Review file · Nature Communications]

Assessing the real implications of CO₂ as generation from renewables increases

Corresponding Author: Professor Ines Azevedo

Version 0:

Reviewer comments:

Reviewer #1

(Remarks to the Author)

Overall assessment: good manuscript with relevant new research on the challenges of high VRE adoption. I personally like that the authors took it one step further: first, estimated the increased emissions of the thermal capacity under low-capacity factors, but also, the net penalty on emissions displaced. I enjoyed their regression formulation. A suggestion in the abstract is to mention that the results come from a regression model. At first, I thought that the paper was only about historical data.

Suggested small changes in the manuscript:

Not so clear how this fits the lit. review as it doesn't mention emissions directly: Other relevant studies include Makarov [24], who studied the operational impact of wind generation in CAISO. They concluded that wind generation has limited effects on load following and regulation within CAISO's operational region.

Table 1 not sure if that's the best placement. It interrupts the reading flow into data that I believe they used. But that is not presented/mentioned in the adjacent pages of the table. The table is neat and clear, it is just not fully integrated with the adjacent text.

We then we use historical  extra "we"

Not clear to me why Fig. 1 justifies the simple scenarios chosen. In the body, the authors state "The motivation for these simple scenarios is explained by Figure 1." Then Fig 1 is explained. But I didn't get clarity of how an evident relationship of capacity factor and emissions intensity justifies the scenarios and steps chosen for the analysis.

In the closing statement " These simple scenarios provide important bounding results: if power plants were to be operating far from their optimal operation limits, i.e., with low capacity factors and high emissions intensity, the overall CO₂ emissions from natural gas and coal power plants in CAISO and ERCOT would be somewhere between 12% and 26% higher than the observed emissions (depending on the year)" I suggest to link this % to what was presented in Fig. 1. Why is the overall increase as high as 26% but the emissions intensity several factors higher at low cap factors? Is it because the plants are simply used very sporadically that even a really high intensity factor doesn't represent a large overall increase? I suggest one additional sentence that aids the reader arriving at this (or the correct) interpretation.

The sentence "Now, we turn to provide more accurate estimates the effect that from renewables on thermal power plants' operations" needs to be reviewed.

An equation for the log. model would be great to understand the model run (it is presented later, but maybe could be added here too?)

(Remarks on code availability)

Will they make their data/model available?

Reviewer #2

(Remarks to the Author)

The research paper analyses the impact of increased generation from wind and solar on the overall CO₂ emissions the CO₂ emissions intensity of the electricity system, considering that individual plants will operate at typically reduced capacity factors, and hence increased CO₂ emissions.

The work is interesting, and important for policy makers. While there have been studies around the subject, the research is significant due to breadth of analysis that is presented in the paper.

The analysis and methodology is sound and the data analysis, conclusions and interpretations are clear. In addition, a substantial amount of useful data is provided in the appendix.

My general comments:

Since the authors are in particularly also addressing policy makers with their study, it would be beneficial to elaborate a bit more on the consequences and potential solutions for the problem they identified (increased emissions on a plant level with increased share of renewables). You could imagine that operating larger plants more flexible would help (in contrast to ramping up and down smaller peaker units). Increased use of energy storage, expansion of transmission lines and demand-side management could also potentially reduce these effects. Also, new thermal plants could be designed more specifically to address the problem of increased emissions at low capacity factors, or generators could be used that have low emission intensities during part-load (e.g. natural gas fired high temperature fuel cells). I think it would be good to address these points, since it shows that increased plant-level at an increased share of variable renewables is not "unavoidable".

It would be good to specify more often, that you are addressing CO₂ emissions. While this is clear in general, it would be good to specify this e.g. in the captions of the figures explicitly, since very different kinds of emissions are associated with power plants (CO, NO_x). Maybe the authors could also comment on how these other emissions would be affected as well (only if the data allow for this, or potentially in a future study).

It is good, that alternative statistical models are provided in section A.4. However, the results obtained for them should be discussed, and it should be clarified what lead to the selection of the final model.

As another general comment, it was for me not helpful to have the methods section at the end of the paper. On p.5 you explain formulas and results, and it is cumbersome to understand and then jump pages to find the equations. I would recommend to restructure the paper and show the methods first.

Here are a few recommendations for updates/calrifications:

In table 2 & 4, the asterix *** and * are not explained.

On p.5, I believe "0.20%" should be corrected to "0.21%".

In Appendix A.1, the numbering system for the different graphs/power plants should be explained (I assume it is a standardized ID used for each power plant from the database, but it should be briefly clarified).

For Figure 3, the axes used should be explained.

In section 4.2.1, the abbreviation ISO should be introduced. It is also spelled differently, and the abbreviation "therm" should be avoided, and "withing" corrected in the sentence:

"Our approach departs from those previous models in that we explicitly control for the total generation of therm power plans within the ISO, total hydro generation within the ISo, net imports, and the generation from wind and from solar." The gap left after the paragraph should be reduced.

Here are a few minor recommendations for changes (typos etc.):

p.3 before "Tolk Station": here seems to be one blank too much

(Remarks on code availability)

Reviewer #3

(Remarks to the Author)

This paper empirically estimates the impact of increases in intermittent renewable power generation on the emissions from dispatchable fossil fuel power generation in California (CAISO) and Texas (ERCOT). Based on a dataset of hourly power generation from 2018-2022, the authors estimate that a 1% increase in solar (wind) power generation results in a 0.21% (0.19%) decrease in total direct emissions from power generation in California and a 0.03% (0.31%) decrease in total direct emissions from power generation in Texas. As such, the authors estimate that solar (wind) power generation in California achieves 91.3% (95.0%) of the emission reductions that would occur if fossil fuel power generators did not suffer from poorer emission performance due to lower capacity utilization and increased ramping.

The analysis in this paper is timely and generally well done. The issue of the reduced cost and emissions performance of fossil fuel power generators due to their increasingly intermittent operation is well known and, as the authors discuss, has been examined in several previous studies. However, an empirical estimate of the magnitude of the effect has been lacking. In addition to these considerations, several key concerns and questions come to mind:

1) Section 2.1: What is the relevance of this section? The analysis here appears somewhat overly simplistic, and Figures 1 and 2 do not seem particularly informative. In particular, the exponential shape of the emission intensity as a function of the capacity factor shown in Figure 1 is probably primarily due to a denominator effect in the sense that some plants exhibit fixed emissions that are divided by low amounts of generation. On this note, I would recommend clarifying the language and making it less sensational (see, for instance, the sentence in the abstract: "We find that the emissions intensity of conventional generation is up to 10x higher at the lower capacity factors that several power plants now operate at.")

2) Section 2.2:

a. This section requires more clarity in the presentation of the method and results. In particular, the text in the main body and the description of Table 2 should explain which control variables and fixed-effects structure have been used to produce the results. Table 2 should also include the fixed effects and a legend for the significance levels. The authors should also provide full regression tables, including all control variables, in the Appendix.

b. Equation 1: How exactly is the variable (in LaTeX writing) \overline{W}_t calculated? This is not fully clear in the description on page 8. Also, why is not simply the statistical variance of W_t used instead? Furthermore, how are zero values in the variables treated in the logarithmic approach?

c. Section 4.2.1: The authors claim that the regression coefficients show a causal change in the dependent variable. This requires further justification and quantitative analysis. It is easy to imagine that, despite the fixed effects used, the error terms are correlated and that the daily solar (or wind) generation is serially correlated. In any case, the authors should provide correlation statistics (in the Appendix) and discuss the findings in the paper (or in Methods).

3) Section 2.3: In this section, the authors present their findings on regressions for individual plants. In particular, the authors run regressions for individual plants and then plot the obtained coefficients in a scatter plot to illustrate the heterogeneity among thermal power generators. This falls somewhat short of what the authors could have done.

a. First, it remains unclear why the authors do not include dummy variables for the different plant types and interactions of each dummy variable with the independent variable of interest (e.g., S_t) in the preceding regression. This should quantify the average effect of the independent variable on a given dependent variable for a particular plant type. This approach not only illustrates but quantifies the heterogeneity across plant types. In terms of fossil fuel plant types, I would suggest differentiating between natural gas combined cycle plants, natural gas peaker plants, and coal power generation. The authors currently only differentiate between natural gas and coal power plants.

b. Second, Figure 3 is a rather unfortunate depiction of the findings. The overlapping colors make it hard to grasp the distribution, and it remains unclear why the authors want to show the geographic dispersion of the plants. A more suitable chart type would be a boxplot. This would quantify the statistical distribution of the coefficients and, as one-dimensional plots, provide space to also show the results for the other dependent variables (emissions and emission intensity). If the authors seek to measure the effect of the proximity of renewable power generation to fossil power generators (which the current figure kind appears to attempt), they should do so by including a corresponding variable in the regression analysis.

c. Third, similar to Figure 3, Figure 4 is also a somewhat unfortunate depiction of the findings. If the authors want to examine the impact of plant size on the results, they should do so quantitatively by including a corresponding variable in the regression. This would quantify the effect rather than illustrate it in a figure.

4) Discussion: I would like to request the authors to discuss, ideally in a quantitative manner, how the results change with (i) higher shares of renewable power generation and (ii) the emergence of storage technologies. Higher shares of renewable power generation could intensify the effect the authors find. Storage technologies, in contrast, could mitigate the effect.

5) Writing: Overall, the paper's writing should be clearer, more concise, and less sensational. There are many places across the manuscript where it is not quite clear what the authors mean, and there are several typos.

6) Appendices:

a. A.4: There seems to be an error in the descriptions of alternative specifications. I think it should be "exclude thermal generation," not "solar generation," as seen in the table. The subsequent descriptions all appear to be one off.

b. A.1: Some plants have higher carbon intensities with higher capacity factors. Could this also drive heterogeneity?

(Remarks on code availability)

I haven't had access to the code.

Reviewer #4

(Remarks to the Author)

(Remarks on code availability)

Version 1:

Reviewer comments:

Reviewer #1

(Remarks to the Author)

The comments were addressed, and I don't have any further comments. I did notice that in the closing statement there is a typo (reads "facts" when it's supposed to read "factors"). Looking forward to reading the published version.

(Remarks on code availability)

Reviewer #2

(Remarks to the Author)

Thank you for considering my feedback in the revised manuscript. For me everything is fine now.

(Remarks on code availability)

Reviewer #3

(Remarks to the Author)

The authors have substantially revised the manuscript and either addressed my comments or explained why doing so is not feasible within the scope of their analysis. Before I can recommend the paper for publication in Nature Communications, I would like to ask them to address the following additional points:

1) The authors have revised the original claim that the regression coefficients show a CAUSAL change in the dependent variable to a statement that the regression coefficients measure the ASSOCIATED change in the dependent variable. Yet, descriptions throughout the manuscript still suggest causality. I suggest that the authors revise the language throughout the manuscript to speak only of correlation. Examples of these descriptions with the misleading language highlighted in capital letters include:

- Abstract: "Overall, the effect is small: we find that solar DISPLACES 92.6% of expected emissions in California, whereas wind DISCPLACES 91.1% of expected emissions in Texas."
- p. 4: "From Table 2, an increase in wind and solar consistently RESULT IN lower thermal generation and lower emissions."
- p. 4: "A 1% increase in generation from solar RESULTS IN a 0.27% reduction in thermal generation and 0.25% decrease in total emissions."
- p. 4: "Looking at the displacement induced from wind, we find that a 1% increase in generation RESULTS IN a 0.34% reduction in thermal generation and 0.31% decrease in total emissions."
- p. 4: "While at first, these numbers may seem insignificant, the hourly ramp in solar generation in CAISO during the day can reach 50%, which would RESULT IN a 1% increase in the emissions intensity of power plants for that hour."
- p. 6: "In contrast to the coefficients for wind and solar generation, an increase in the variability of the wind RESULTS IN a subsequent increase in generation from thermal plants in both regions." [Note also that the "the" before "wind results in ..." appears superfluous.]
- p. 6: "A positive coefficient implies that the dependent variable increases IN RESPONSE TO generation from renewables and vice versa. We find that natural gas plants in CAISO show a very heterogeneous RESPONSE TO generation from solar with plants RESPONDING via an increase as well as a decrease in generation, emissions, and emissions intensity. In contrast, all but two plants in ERCOT lower their generation and emissions IN RESPONSE TO increasing generation from wind. However, the emissions intensity of all coal plants increases IN RESPONSE TO a marginal increase in wind generation, and the same is true for most gas plants. [...] The heterogeneous RESPONSE of natural gas plants in CAISO as compared to ERCOT may also be attributed to a shift in generation patterns as more utility-scale solar comes online."
- p. 6: "All plants are natural gas-fired in CAISO, and a mix of natural gas and coal in ERCOT. In CAISO, larger plants lower their generation IN RESPONSE TO increased generation from solar, although the magnitude of this deviation is smaller compared to plants with a lower nameplate capacity. [...] The magnitude of their RESPONSE with increase generation from wind is not pronounced, with larger plants not deviating beyond -0.2."
- p. 8: "At a systems-level, we find that renewables DISPLACE thermal generation and emissions in both regions, although the magnitude of the DISPLACEMENT varies by the renewable resource."
- p. 8: "In addition to the panel regression model that describes plant RESPONSE for the fleet [...] Overall, plants with a smaller nameplate capacity show a wider envelope of solar and wind coefficients IN RESPONSE TO increasing renewable generation."
- p. 8: "In summary, we show that the first order EFFECT of increasing generation from renewables IS lowered emissions and generation from thermal power plants."

2) On p. 4, the authors write: "The model accounts for location and time fixed effects which allows us to interpret the

coefficients independently from the plant fuel type.” I suggest replacing “location” with “plant” and ensuring consistent use of the term “plant fixed-effects” throughout the manuscript. The term “location” leaves unnecessary room for interpretation and could refer to coarser fixed effects than plant fixed-effects.

3) The description in Table 2 includes the time fixed-effects but not the plant fixed-effects. But, as far as I understand, the regression results shown in Table 2 include plant fixed-effects. So I would suggest including them in the description as well.

4) In response to one of my initial comments, the authors provided an explanation of why they used their own measure of wind intermittency instead of simply the variance of wind generation in the analysis. It seems helpful to readers to also include a brief explanation of this when they introduce their measure in equation (2) in Methods.

(Remarks on code availability)

Reviewer #4

(Remarks to the Author)

(Remarks on code availability)

Version 2:

Reviewer comments:

Reviewer #3

(Remarks to the Author)

The authors have addressed all my remaining comments. I congratulate the authors on this paper and recommend it for publication in Nature Communications.

(Remarks on code availability)

Reviewer #4

(Remarks to the Author)

(Remarks on code availability)

The impact of renewables on thermal power plants - reviewer responses. v1 - 09/03

We thank all reviewers for taking out the time to critically review the manuscript. Reviewer comments are highlighted in blue whereas our responses to the comments are formatted in regular text.

In the revised version of the manuscript, all modifications are in red.

Reviewer 1

R1: “Overall assessment: good manuscript with relevant new research on the challenges of high VRE adoption. I personally like that the authors took it one step further: first, estimated the increased emissions of the thermal capacity under low-capacity factors, but also, the net penalty on emissions displaced. I enjoyed their regression formulation.

A suggestion in the abstract is to mention that the results come from a regression model. At first, I thought that the paper was only about historical data.”

Response from authors: The abstract now reads as follows (updated text is underlined): “We use hourly data at the power plant level for the California Independent System Operator and for the Electricity Reliability Council of Texas to estimate the magnitude of the emissions’ penalty associated with renewable variability using a fixed effects regression formulation.”

R1: “Suggested small changes in the manuscript:

Not so clear how this fits the lit. review as it doesn't mention emissions directly: Other relevant studies include Makarov [24], who studied the operational impact of wind generation in CAISO. They concluded that wind generation has limited effects on load following and regulation within CAISO's operational region.”

Response from authors: While we understand the reviewer's comment that this paper does not fit in our literature review since it does not include emissions, we still believe Makarov's paper has an important and related contribution. Makarov introduced the concept of load-following plants, wherein certain power plants in a thermal fleet follow changes and variability in load or demand at the balancing authority level.

In our case, based on the coefficients of the individual power plant regression model, we observe considerable heterogeneity in the response of thermal power plants, based on their nameplate capacity, annual capacity factor and fuel. Makarov's analysis on the operational impact of wind on thermal power plants was published in 2009, when there was very limited historical data on CAISO's market operation in response to increasing generation from wind.

Our analysis builds on this by quantifying the displacement effect based on observed generations and emissions data.

The first paragraph of the literature review has now been modified to read as follows:

“Interest in how variable RES could affect thermal generators proliferated as soon as it became apparent that RES penetration would grow. Earlier studies lacked historical datasets like the one we use in this paper, and relied heavily on simulations to predict operational impacts of RES penetration. For example, Makarov et al. \cite{makarov2009operational} used a simulation model that predicted that CAISO would require additional load following capacity (0.8 GW up, 0.6 GW down) to integrate 4.1 GW of new wind generation from 2006 to 2010. They further showed that load following plants wherein certain power plants in a thermal fleet *follow* changes and variability in load or demand at the balancing-authority level. Katzenstein and Apt \cite{katzenstein2009air} used data from nine natural gas turbines of two types to build a model of a wind or solar photovoltaic plus gas system and find that steam injection gas generators achieve only 30-50\% of expected NO_x emissions reductions and close to 80\% of expected CO_2 emissions reductions.”

Table 1 not sure if that's the best placement. It interrupts the reading flow into data that I believe they used. But that is not presented/mentioned in the adjacent pages of the table. The table is neat and clear, it is just not fully integrated with the adjacent text.

Response from authors: The positioning of the table will be readjusted in the published version (this is one of our OverLeaf struggles...).

We then we use historical  extra "we"

Response from authors: Typo corrected.

Not clear to me why Fig. 1 justifies the simple scenarios chosen. In the body, the authors state "The motivation for these simple scenarios is explained by Figure 1." Then Fig 1 is explained. But I didn't get clarity of how an evident relationship of capacity factor and emissions intensity justifies the scenarios and steps chosen for the analysis.

Response from authors: The second paragraph of Section 2.1 referencing the figure has been updated to the following (additions have been underlined):

The motivation for these simple scenarios is explained by Figure \ref{fig:sample_CF}, which illustrates the relationship between emissions intensity and capacity factor by generating unit for two thermal plants in CAISO and ERCOT. Operating the two sample plants at a capacity factor of 0.05 as opposed to 0.3 increases the emissions intensity of the plants by 133\% and 51\%, respectively. Using CEMS data, we have observed across all thermal plants that the emissions intensity changes markedly as the capacity factor changes. At lower capacity factors, plants consume more fuel per

unit of electricity generated and thus have a higher emissions intensity. This is because they are not operating at their ideal operational levels. Each data point in the figure represents an hourly operational value at which the generation and emissions were measured. In both instances, we see that the emissions intensity at lower capacity factors is several multiples of the baseline emissions intensity at higher utilization. For Moss Landing, a natural gas plant in CAISO, the mean emissions intensity for a capacity factor greater than 0.3 is 0.36 ton CO_2 /MWh. In contrast, when the capacity factor is less than 0.05, the emissions intensity of the plant is 0.84 ton CO_2 /MWh. Tolk Station, a coal-fired plant in ERCOT, generally serves as a baseload plant. The mean emissions intensity for a capacity factor greater than 0.3 is 0.94 ton CO_2 /MWh whereas when the capacity factor is less than 0.05, the mean emissions intensity is 1.426 ton CO_2 /MWh. This increase in emissions is not unique to the two plants in question, but is seen across the entire fleet of thermal plants in CAISO and ERCOT. In the SI, Section \ref{EI_CF_plots}, we show similar plots for all natural gas and coal plants in CAISO and ERCOT.

Furthermore, following this suggestion, Figure 2 has been modified to include a boxplot of the mean, P10, and P90 emissions intensities across all plants for a given year.

R1: "In the closing statement " These simple scenarios provide important bounding results: if power plants were to be operating far from their optimal operation limits, i.e., with low capacity factors and high emissions intensity, the overall CO2 emissions from natural gas and coal power plants in CAISO and ERCOT would be somewhere between 12% and 26% higher than the observed emissions (depending on the year)" I suggest to link this % to what was presented in Fig. 1. Why is the overall increase as high as 26% but the emissions intensity several factors higher at low capacity factors? Is it because the plants are simply used very sporadically that even a really high intensity factor doesn't represent a large overall increase? I suggest one additional sentence that aids the reader arriving at this (or the correct) interpretation."

Response from authors: The closing statement has been amended as follows:

"These simple scenarios provide important bounding results: if power plants were to be operating far from their optimal operation limits, i.e., with low capacity factors and high emissions intensity, the overall CO2 emissions from natural gas and coal power plants in CAISO and ERCOT would be somewhere between 12% and 26% higher than the observed emissions (depending on the year). Although the emissions intensity at low capacity factor operation is several factors higher than the baseline emissions intensity, plants are operated infrequently at very low capacity factors. Hence, the 90th percentile of emissions intensity is lower than the extreme values observed in Figure 1."

R1: "The sentence "Now, we turn to provide more accurate estimates the effect that from renewables on thermal power plants' operations" needs to be reviewed".

Response from authors: This sentence has been amended to read as follows: “Now, we turn to provide more accurate estimates on the effect of renewables on thermal power plant operations, using both entity and time fixed-effects panel regression model with a logarithmic specification.

R1: “An equation for the log. model would be great to understand the model run (it is presented later, but maybe could be added here too?)”

Response from authors: We received a similar recommendation from Reviewer 3 to include more information about the fixed-effects model and key variables used for readers. The first paragraph of this section has been modified to read as follows:

“Now, we turn to provide more accurate estimates on the effect of renewables on thermal power plant operations, using both entity and time fixed-effects panel regression models with a logarithmic specification. This model describes the change in the dependent variable (generation, emissions, or emissions intensity) as a function of the generation from wind and solar, residual thermal demand, individual power-plant entity effects, time effects, and other effects from control variables. The logarithmic specification used is given by Equation \ref{log_spec_1}. The control variables include the sum of wind generation, solar generation, and demand of the trading partners of the ISO and the intermittency of wind generation average over the course of the day. Intermittency in wind generation refers to the variability in power output over time, which is quantified by assessing the fluctuations between consecutive hourly wind generation levels. This is calculated by first determining the hourly differences in wind generation values, and subsequently summing the absolute magnitudes of these differences. Instead of considering generation from renewables as a whole, we separate solar and wind generation as discrete independent variables given their varying levels of penetration in CAISO and ERCOT. Time fixed-effects control for the month and year of daily observations. The coefficients of this model can thus be interpreted as the percent change in the dependent variable (emissions intensity, total emissions and generation), per unit percent increase in generation from wind and solar. The model accounts for location and time fixed effects which allows us to interpret the coefficients independently from the plant fuel type.

Reviewer 2

R2: “The research paper analyses the impact of increased generation from wind and solar on the overall CO2 emissions the CO2 emissions intensity of the electricity system, considering that individual plants will operate at typically reduced capacity factors, and hence increased CO2 emissions.

The work is interesting, and important for policy makers. While there have been studies around the subject, the research is significant due to breadth of analysis that is presented in the paper.

The analysis and methodology is sound and the data analysis, conclusions and interpretations are clear. In addition, a substantial amount of useful data is provided in the appendix.

My general comments:

Since the authors are in particular also addressing policy makers with their study, it would be beneficial to elaborate a bit more on the consequences and potential solutions for the problem they identified (increased emissions on a plant level with increased share of renewables). You could imagine that operating larger plants more flexible would help (in contrast to ramping up and down smaller peaker units). Increased use of energy storage, expansion of transmission lines and demand-side management could also potentially reduce these effects. Also, new thermal plants could be designed more specifically to address the problem of increased emissions at low capacity factors, or generators could be used that have low emission intensities during part-load (e.g. natural gas fired high temperature fuel cells). I think it would be good to address these points, since it shows that increased plant-level at an increased share of variable renewables is not "unavoidable".

Response from authors: These are great points, which we have now incorporated in the last paragraph of the discussion section:

"In summary, we show that the first order effect of increasing generation from renewables is lowered emissions and generation from thermal power plants. However, the displacement is not 1:1 as expected. In fact, smaller peaker plants are more prone to ramping up and down at a lower capacity factor and higher heat rate, resulting in higher instantaneous emissions and emissions intensity. As more renewables are installed in the grid, we expect that traditionally baseload plants will be increasingly made to cycle in a manner similar to the peaker plants, leading to a further inefficiency at the plant level. Policymakers may want to consider different strategies to mitigate this effect and to achieve a combined thermal displacement and emissions reduction goal. While it is outside the scope of this paper to model the implications of different strategies, some strategies may warrant attention in future analysis. For example, storage technologies could be used as a means to sustain constant capacity factors and avoid ramping, and could regulate and even substitute the use of spinning reserves. Similarly, there may be emissions reduction benefits in ramping a few bigger units rather than cycling more frequently a large number of small peaking units. Other strategies may include the expansion of new transmission infrastructure or demand-side management strategies such as demand response and targeted building-level energy efficiency retrofits. For supply-side infrastructure, thermal power plants can be designed to operate efficiently at low capacity factors or part-load operations by improving the efficiency of boilers and steam turbines with variable steam flow rate, or by incorporating natural gas fired high temperature fuel cells. Addressing these secondary impacts is critical for consistently lowering emissions in a low-carbon future.

R1: "It would be good to specify more often, that you are addressing CO2 emissions. While this is clear in general, it would be good to specify this e.g. in the captions of the figures explicitly, since very different kinds of emissions are associated with power plants (CO, NOx). Maybe the authors

could also comment on how these other emissions would be affected as well (only if the data allow for this, or potentially in a future study).”

Response from authors: In all figures, we have now specifically stated that the emissions refer to CO₂ emissions.

Given the complexity of the models and heterogeneity in plant response across other gasses (SO₂ and NO_x) we will be considering this suggestion as the subject of a subsequent study!

R2: “It is good, that alternative statistical models are provided in section A.4. However, the results obtained for them should be discussed, and it should be clarified what lead to the selection of the final model.”

Response from authors: A detailed explanation of alternate statistical models and the criteria for the main specification’s selection has been added to the SI. Section C3

R2: “As another general comment, it was for me not helpful to have the methods section at the end of the paper. On p.5 you explain formulas and results, and it is cumbersome to understand and then jump pages to find the equations. I would recommend to restructure the paper and show the methods first.”

Response from authors: While we understand this concern, this is a general formatting requirement for Nature Communications publications. Where possible, we have provided an overview of the methods and interpretation of coefficients prior to discussing the results in each subsection.

R2: “Here are a few recommendations for updates/clarifications:

In table 2 & 4, the asterix *** and * are not explained.”

Response from authors: This has been corrected.

R2: “On p.5, I believe “0.20%” should be corrected to “0.21%”.”

Response from authors: This has been corrected.

R2: “In Appendix A.1, the numbering system for the different graphs/power plants should be explained (I assume it is a standardized ID used for each power plant from the database, but it should be briefly clarified).”

Response from authors: Good point: the numbers represent the ORISPL number of each power plant. This has now been mentioned in the SI. The description for the figures now reads as follows: Figures \ref{fig:hourly_metrics-2} through \ref{fig:hourly_metrics-10} show the variation of emissions intensity as a function of capacity factor for thermal plants in CAISO and ERCOT. Plants are numbered by their unique ORISPL IDs.

R2: "For Figure 3, the axes used should be explained."

Response from authors: The original Figure 3 has now been removed. In place of a map showing the geographic distribution of plants, we now present power plant coefficients as a box plot.

R2: "In section 4.2.1, the abbreviation ISO should be introduced. It is also spelled differently, and the abbreviation "therm" should be avoided, and "withing" corrected in the sentence: "Our approach departs from those previous models in that we explicitly control for the total generation of therm power plants within the ISO, total hydro generation within the ISO, net imports, and the generation from wind and from solar." The gap left after the paragraph should be reduced."

Response from authors: Done.

R2: "Here are a few minor recommendations for changes (typos etc.):
p.3 before "Tolk Station": here seems to be one blank too much"

Response from authors: Done.

Reviewer 3

R3: "This paper empirically estimates the impact of increases in intermittent renewable power generation on the emissions from dispatchable fossil fuel power generation in California (CAISO) and Texas (ERCOT). Based on a dataset of hourly power generation from 2018-2022, the authors estimate that a 1% increase in solar (wind) power generation results in a 0.21% (0.19%) decrease in total direct emissions from power generation in California and a 0.03% (0.31%) decrease in total direct emissions from power generation in Texas. As such, the authors estimate that solar (wind) power generation in California achieves 91.3% (95.0%) of the emission reductions that would occur if fossil fuel power generators did not suffer from poorer emission performance due to lower capacity utilization and increased ramping.

The analysis in this paper is timely and generally well done. The issue of the reduced cost and emissions performance of fossil fuel power generators due to their increasingly intermittent operation is well known and, as the authors discuss, has been examined in several previous studies. However, an empirical estimate of the magnitude of the effect has been lacking. In addition to these considerations, several key concerns and questions come to mind:

1) Section 2.1: What is the relevance of this section? The analysis here appears somewhat overly simplistic, and Figures 1 and 2 do not seem particularly informative. In particular, the exponential shape of the emission intensity as a function of the capacity factor shown in Figure 1 is probably primarily due to a denominator effect in the sense that some plants exhibit fixed emissions that are divided by low amounts of generation. On this note, I would recommend clarifying the language and making it less sensational (see, for instance, the sentence in the abstract: "We find that the emissions intensity of conventional generation is up to 10x higher at the lower capacity factors that several power plants now operate at.")".

Response from authors: Thank you for this comment. The critical value of Section 2.1 is to highlight the magnitude of emissions intensity for a few actual power plants as a function of the capacity factor. This sets the stage for the analysis shown in Section 2.2, and motivates the importance of the regression analysis shown in Section 2.3. We also see that this behavior occurs in many other power plants in the SI. Importantly, we observe that emissions intensity is constant and does not hold a linear pattern as generation increases. We would like to keep this figure and associated text given this explanation, but we also want to address the reviewer concerns. Thus, we have now revised the figure shown below. This figure shows the annual emissions on the left when computed under different assumptions (this is our old Figure 3). In the panel on the right, we show the distribution of P10, mean, and P90 annual emissions intensities across all power plants of the same fuel type in CAISO (top) and in ERCOT (bottom). It's clear that there is substantial heterogeneity amongst plants of the same fuel type, which we then quantify in the following section.

As suggested by the reviewer, we have reworked the abstract and the language across the text as we definitely do not want to overclaim any of our results.

2) Section 2.2:

a. This section requires more clarity in the presentation of the method and results. In particular, the text in the main body and the description of Table 2 should explain which control variables and fixed-effects structure have been used to produce the results. Table 2 should also include the fixed effects and a legend for the significance levels. The authors should also provide full regression tables, including all control variables, in the Appendix.

Response from authors: We understand the frustration from the reviewer, but we would like to clarify that Nature Communications requires the methods to be at the end of the manuscript (Section 4 in our paper). In that section, we include both the full model specification and define each of the control variables. We have provided a now more detail in the text of Section 2.2 as follows:

- The fixed-effects formulation used is both an entity and time fixed-effects structure
- Control variables include the intermittency in wind generation averaged over each day, and the sum of wind generation, solar generation and demand for the trading partners of the ISO to which the plant belongs. This description has been included in the main text and the caption of Table 2
- A legend for significance levels has also been included in the caption of Table 2.

Specifically, the text reads:

“Now, we turn to provide more estimates on the effect of renewables on thermal power plant operations using entity and time as fixed-effects in a panel regression model with a logarithmic specification. This model describes the change in the dependent variable (generation, emissions, or emissions intensity) as a function of the generation from wind and solar, residual thermal demand, individual power-plant entity effects, time effects, and other effects from control variables. The logarithmic specification used is given by Equation $\ref{log_spec_1}$. The control variables are the sum of wind generation, solar generation, and demand of the trading partners of the ISO; and the intermittency of wind generation average for the day. Intermittency in wind generation refers to the variability in power output over time, quantified by assessing the fluctuations between consecutive hourly wind generation levels. This is calculated by (i) determining the hourly differences in wind generation values and (ii) summing the absolute magnitudes of these differences. Instead of considering generation from renewables as a whole, we separate solar and wind generation as discrete independent variables given their varying penetration levels in CAISO and ERCOT. Time fixed-effects control for the month and year of daily observations. The coefficients of this model can thus be interpreted as the percent change in the dependent variable (emissions intensity, total emissions, and generation), per unit percent increase in generation from wind and solar. The model accounts for location and time fixed effects, allowing us to interpret the coefficients independently from the plant fuel type. More details regarding the methods and data are provided in Section 4.”

To address the reviewer's comment on providing the coefficients for all variables, including the fixed-effects, we only omit the entity fixed-effects terms and include all other coefficients in the table.

b. Equation 1: How exactly is the variable (in LaTeX writing) \overline{W}_t calculated? This is not fully clear in the description on page 8. Also, why is not simply the statistical variance of W_t used instead? Furthermore, how are zero values in the variables treated in the logarithmic approach?

Response from authors: An equation for \overline{W}_t has now been included (and shown below).

$$\overline{W}_t = \sum_{h=0}^{23} |W_{t,h+1} - W_{t,h}|$$

Although variance for wind over a given day can be used in place of \overline{W}_t , the interpretation of the coefficient will be different. Differences between the two are as follows:

- Nature of wind variability
 - Intermittency (present approach) focuses on fluctuations or changes between consecutive hours.
 - Variance captures overall daily deviations from the average wind generation, regardless of the specific timing of the fluctuations
- Impact on grid stability and the generation mix
 - Intermittency would suggest that frequent short-term fluctuations in wind generation lead to operational challenges for balancing generation, potentially increasing the need for backup generation (hence more emissions).
 - Variance would suggest that higher dispersion in wind generation over the course of a day impacts grid operations and emissions, but without focusing on the timing of fluctuations. It emphasizes how much wind generation deviates from its daily mean rather than the sequence of deviations.

Zero values occur most frequently in the hourly dataset (given the diurnal pattern of solar generation) and not in the aggregated daily dataset. Where zero values occur, we neglect that particular row.

c. Section 4.2.1: The authors claim that the regression coefficients show a causal change in the dependent variable. This requires further justification and quantitative analysis. It is easy to imagine that, despite the fixed effects used, the error terms are correlated and that the daily solar (or wind) generation is serially correlated. In any case, the authors should provide correlation statistics (in the Appendix) and discuss the findings in the paper (or in Methods).

Response from authors: Correlation statistics have now been included in the SI. We have also presented the results of the Durbin-Watson test for autocorrelation in the residuals.

“To test for correlation between each thermal power plant series and other exogenous variables, we conduct several statistical tests. Figure \ref{fig:corr1} shows the Pearson correlation coefficient for all exogenous variables. Wind and solar generation in a given ISO are partially correlated with corresponding generation from the ISO's trading partners, however, the correlation is not pronounced.”

“We use the Durbin-Watson (DW) test \cite{white1992durbin} to check for autocorrelation in the residuals of the panel regression model for each entity. A value close to 2 suggests no autocorrelation, while values closer to 0 or 4 indicate positive or negative autocorrelation, respectively. 73\% of thermal power plants have a DW test statistic between 1 and 2, indicating that there is very limited autocorrelation in the residuals of the panel data.”

Finally, to ensure clarity, we have replaced “causation” with “association”.

3) Section 2.3: In this section, the authors present their findings on regressions for individual plants. In particular, the authors run regressions for individual plants and then plot the obtained coefficients in a scatter plot to illustrate the heterogeneity among thermal power generators. This falls somewhat short of what the authors could have done.

a. First, it remains unclear why the authors do not include dummy variables for the different plant types and interactions of each dummy variable with the independent variable of interest (e.g., $\$S_t\$$) in the preceding regression. This should quantify the average effect of the independent variable on a given dependent variable for a particular plant type. This approach not only illustrates but quantifies the heterogeneity across plant types. In terms of fossil fuel plant types, I would suggest differentiating between natural gas combined cycle plants, natural gas peaker plants, and coal power generation. The authors currently only differentiate between natural gas and coal power plants.

Response from authors: Including plant-type interactions is indeed plausible, however limits the interpretation especially given that plants of the same fuel type respond differently to variable generation from solar and wind. This can be mitigated by interacting plant ID with solar and wind, however, doing so results in an over-specified model with too many coefficients and interaction terms.

In the previous panel regression, we control for entity-fixed effects and hence already implicitly account for heterogeneity in plant-type, whereas the main goal of Section 2.3 is to further analyze plant-level changes in generation and emissions rather than the technology or fuel.

The reviewer’s suggestion regarding differentiating between natural gas combined cycle plants, peaker plants, and coal power generation is great. We have pursued that additional analysis and are including the results. We summarize the key findings in the main manuscript as follows:

“Figure \ref{fig:ols_coefs} shows the distribution of coefficients for plant generation, emissions, and emissions intensity in response to solar in CAISO (left) and wind in

ERCOT (right). A positive coefficient implies that the dependent variable increases in response to generation from renewables and vice versa. We find that natural gas plants in CAISO show a very heterogeneous response to generation from solar with plants responding via an increase as well as a decrease in generation, emissions, and emissions intensity. In contrast, all but two plants in ERCOT lower their generation and emissions in response to increasing generation from wind. However, the emissions intensity of all coal plants increases in response to a marginal increase in wind generation, and the same is true for most gas plants.

The heterogeneous response of natural gas plants in CAISO as compared to ERCOT may also be attributed to a shift in generation patterns as more utility-scale solar comes online. Fast-ramping peaker plants replace base load combined cycle plants during peak production hours. In CAISO, 4 out of 7 plants with a coefficient greater than 1.0 account for 28% of generation and also exhibit the highest emissions intensity of all plants in the balancing region. In ERCOT, plant behavior is very different to that of CAISO. The mean coefficient for generation is -0.41, which is close to our fleet level coefficient of -0.34 from the panel regression. Coal plants with a relatively smaller nameplate capacity are characterized by a higher emissions intensity, even though the capacity factor of these plants is comparable to that of larger plants. The two smallest coal plants in ERCOT show the largest deviation in response to wind (-2.25%, -2.07%) and also have a NO_x emissions intensity 60% higher than the mean of all coal plants indicating more frequent ramping and startup.”

b. Second, Figure 3 is a rather unfortunate depiction of the findings. The overlapping colors make it hard to grasp the distribution, and it remains unclear why the authors want to show the geographic dispersion of the plants. A more suitable chart type would be a boxplot. This would quantify the statistical distribution of the coefficients and, as one-dimensional plots, provide space to also show the results for the other dependent variables (emissions and emission intensity). If the authors seek to measure the effect of the proximity of renewable power generation to fossil power generators (which the current figure kind appears to attempt), they should do so by including a corresponding variable in the regression analysis.

Response from authors: Good point. In the original manuscript, Figure 3 allowed readers to observe heterogeneity in power plant coefficients across thermal power plants in two diverse electric power systems. More than the interpretation and distribution of the coefficients, including this Figure allowed readers to visually observe the variability in thermal plant response across a region with a high penetration of solar (CAISO) vs wind (ERCOT).

We have now included a box plot of individual coefficients across generation, emissions, and emissions intensity by fuel type in CAISO and ERCOT as a main figure. Indeed, this captures plant heterogeneity’s effect much better for interpretation - thank you for the suggestion.

We are not seeking to measure the effect of proximity to renewable generation in the present study, although this could be included in a forthcoming analysis.

c. Third, similar to Figure 3, Figure 4 is also a somewhat unfortunate depiction of the findings. If the authors want to examine the impact of plant size on the results, they should do so quantitatively by including a corresponding variable in the regression. This would quantify the effect rather than illustrate it in a figure.

Response from authors: Figure 4 is generated by interpreting the coefficients of the ordinary least squares specification applied to individual thermal plants. If it were a panel regression formulation considering a panel of all thermal plants, it would be feasible to include the nameplate capacity as an independent variable, however in our case, since we individually evaluate the coefficients based on the hourly data of every plant, the proposed alternate method is not viable. Furthermore, another reason for individual OLS models on hourly data is to estimate the expected vs actual emissions displacement introduced by Graf et al. In Figures 15a and 15b in the SI, we quantify the magnitude of the expected emissions displacement of thermal plants. Figure 4 is used to explore a cross-section of multiple variables without the risk of over-specifying the regression model.

Additionally, given the heterogeneity in power plant operations depending on the intricacies of unit commitment and economic dispatch, plants with a similar nameplate capacity are dissimilar in their annual emissions intensity, and renewable response. Figure 4 is used to clearly illustrate this in supplement to the updated boxplot in Figure 3.

4) Discussion: I would like to request the authors to discuss, ideally in a quantitative manner, how the results change with (i) higher shares of renewable power generation and (ii) the emergence of storage technologies. Higher shares of renewable power generation could intensify the effect the authors find. Storage technologies, in contrast, could mitigate the effect.

Response from authors: While we agree with the reviewer that quantifying power plant emissions and operational behavior in a future system with more renewable generation or storage technologies is a fundamental question, doing so would require the use of a capacity expansion and dispatch models, assigning heat rate and emissions intensity curves to each generating unit, and then analyzing system-level emissions under different renewable penetration scenarios. This is out of scope of the present analysis. However, in this current paper, we look to quantify the magnitude of the displacement effect, and find whether power plants are indeed becoming more polluting using observed historical data.

Storage technologies can indeed mitigate this effect. Although it is not possible to quantify by how much, we have modified the discussion to reflect this.

In addition, the commentary in the results section addresses this comment on quantifying how results change with higher shares of renewables power. For instance:

“While the expected vs actual emissions displacement helps us understand the system-level impact of renewables, it still does not attribute the effect of renewables on plant-level emissions intensity. Looking closely at the coefficients of the model with emissions intensity as the dependent variable, we find that a 1% increased generation from wind and solar does indeed impact plant-level emissions intensity for

existing thermal generators. For CAISO, thermal power plant intensities are more susceptible to increased generation from solar as compared to wind (0.02% for solar, 0.01% for wind). In ERCOT, generation from wind results in a 0.02% increase in emissions intensity of plants. While at first, these numbers may seem insignificant, the hourly ramp in solar generation in CAISO during the day can reach ~50%, which would result in a 1% increase in the emissions intensity of power plants for that hour. If we consider CAISO's 2022 system-level emissions averaged uniformly across all hours, a 1% increase in intensity across all power plants is equivalent to 53 tons of additional CO_2 for that given hour. As the ramp becomes more pronounced with increasing intermittent renewable capacity, this marginal increase should not be ignored. Meeting 2030 and 2045 clean energy targets in both CAISO and ERCOT will require the installation of more installed renewable capacity, resulting in an even higher percent change in generation from renewables across all days of the year. Figure S5 in the SI demonstrates the change in CO_2 emissions intensity for a 0-100% increase in generation from solar and wind in CAISO and ERCOT."

5) Writing: Overall, the paper's writing should be clearer, more concise, and less sensational. There are many places across the manuscript where it is not quite clear what the authors mean, and there are several typos.

Response from authors: The typos have been corrected and the language has been amended to be less sensational and concise.

6) Appendices:

a. A.4: There seems to be an error in the descriptions of alternative specifications. I think it should be "exclude thermal generation," not "solar generation," as seen in the table. The subsequent descriptions all appear to be one off.

Response from authors: The order of equations has been amended.

b. A.1: Some plants have higher carbon intensities with higher capacity factors. Could this also drive heterogeneity?

Response from authors: Indeed. This is especially pronounced for power plants that have multiple units, although the share of these plants is minimal and hence not a driving effect.

The impact of renewables on thermal power plants - reviewer responses. v1 - 01/08

We thank all reviewers for taking out the time to critically review the manuscript. We appreciate the feedback that you have provided thus far. It goes a long way in allowing us to improve the quality of our work.

In this document, reviewer comments are highlighted in blue whereas our responses to the comments are formatted in regular text.

In the revised version of the manuscript, all modifications are in blue.

Reviewer #1 (Remarks to the Author):

The comments were addressed, and I don't have any further comments. I did notice that in the closing statement there is a typo (reads "facts" when it's supposed to read "factors"). Looking forward to reading the published version.

Response from authors: Thank you for your comments and feedback. The typo has been corrected.

Reviewer #2 (Remarks to the Author):

Thank you for considering my feedback in the revised manuscript. For me everything is fine now.

Response from authors: Thank you for your comments and feedback.

Reviewer #3 (Remarks to the Author):

The authors have substantially revised the manuscript and either addressed my comments or explained why doing so is not feasible within the scope of their analysis. Before I can recommend the paper for publication in Nature Communications, I would like to ask them to address the following additional points:

1) The authors have revised the original claim that the regression coefficients show a CAUSAL change in the dependent variable to a statement that the regression coefficients measure the ASSOCIATED change in the dependent variable. Yet, descriptions throughout the manuscript still suggest causality. I suggest that the authors revise the language throughout the manuscript to speak

only of correlation. Examples of these descriptions with the misleading language highlighted in capital letters include:

Response from authors: thank you for this suggestion. Where applicable, we have modified the text in the manuscript to suggest correlation / association as opposed to a causal change. Specific instances mentioned below include their corrected language which is also reflected in the revision.

- Abstract: “Overall, the effect is small: we find that solar DISPLACES 92.6% of expected emissions in California, whereas wind DISPLACES 91.1% of expected emissions in Texas.”

Response from authors: this sentence now reads as follows: “Overall, the relationship is modest: our analysis indicates that solar generation is associated with a 92.6% reduction in expected emissions in California, while wind generation is associated with a 91.1% reduction in expected emissions in Texas.”

- p. 4: “From Table 2, an increase in wind and solar consistently RESULT IN lower thermal generation and lower emissions.”

Response from authors: this sentence now reads as follows: “From Table 2, an increase in wind and solar generation is associated with lower thermal generation and emissions.”

- p. 4: “A 1% increase in generation from solar RESULTS IN a 0.27% reduction in thermal generation and 0.25% decrease in total emissions.”

- Response from authors: this sentence now reads as follows: “Our regression estimates indicate that a 1% increase in solar generation is associated with a 0.27% decrease in thermal generation and a 0.25% decrease in total emissions.”

- p. 4: “Looking at the displacement induced from wind, we find that a 1% increase in generation RESULTS IN a 0.34% reduction in thermal generation and 0.31% decrease in total emissions.”

Response from authors: this sentence now reads as follows: “Examining the relationship between wind generation and displacement, our analysis shows that a 1% increase in wind generation is associated with a 0.34% decrease in thermal generation and a 0.31% decrease in total emissions.”

- p. 4: “While at first, these numbers may seem insignificant, the hourly ramp in solar generation in CAISO during the day can reach 50%, which would RESULT IN a 1% increase in the emissions intensity of power plants for that hour.”

Response from authors: this sentence now reads as follows: “While these coefficients may initially appear small, the hourly ramp in solar generation in CAISO

during the day can reach 50%, which corresponds to an estimated 1% increase in the emissions intensity of power plants for that hour, according to our model.”

- p. 6: “In contrast to the coefficients for wind and solar generation, an increase in the variability of the wind RESULTS IN a subsequent increase in generation from thermal plants in both regions.” [Note also that the “the” before “wind results in ...” appears superfluous.]

Response from authors: this sentence now reads as follows: “In contrast to the coefficients for wind and solar generation, higher wind variability is associated with higher generation from thermal plants in both regions.”

- p. 6: “A positive coefficient implies that the dependent variable increases IN RESPONSE TO generation from renewables and vice versa. We find that natural gas plants in CAISO show a very heterogeneous RESPONSE TO generation from solar with plants RESPONDING via an increase as well as a decrease in generation, emissions, and emissions intensity. In contrast, all but two plants in ERCOT lower their generation and emissions IN RESPONSE TO increasing generation from wind. However, the emissions intensity of all coal plants increases IN RESPONSE TO a marginal increase in wind generation, and the same is true for most gas plants. [...] The heterogeneous RESPONSE of natural gas plants in CAISO as compared to ERCOT may also be attributed to a shift in generation patterns as more utility-scale solar comes online.”

Response from authors: this sentence now reads as follows: “A positive coefficient indicates a positive correlation between the dependent variable and renewable generation, and vice versa. We find that natural gas plants in CAISO show highly heterogeneous relationships with solar generation, and the associated change in generation, emissions, and emissions intensity. In contrast, all but two plants in ERCOT demonstrate lower generation and emissions with higher wind generation. However, the emissions intensity of all coal plants and most gas plants shows positive correlation with marginal increases in wind generation.”

The heterogeneous relationship between natural gas plants in CAISO compared to ERCOT may be related to shifting generation patterns accompanying increased utility-scale solar capacity.

- p. 6: “All plants are natural gas-fired in CAISO, and a mix of natural gas and coal in ERCOT. In CAISO, larger plants lower their generation IN RESPONSE TO increased generation from solar, although the magnitude of this deviation is smaller compared to plants with a lower nameplate capacity. [...] The magnitude of their RESPONSE with increase generation from wind is not pronounced, with larger plants not deviating beyond -0.2.”

Response from authors: this sentence now reads as follows: “In CAISO, larger plants exhibit negative correlations with solar generation, although the magnitude of this correlation is smaller compared to plants with lower nameplate capacity.”

The coefficients associated with wind generation show less variation for larger plants, with values not falling below -0.2.

- p. 8: “At a systems-level, we find that renewables DISPLACE thermal generation and emissions in both regions, although the magnitude of the DISPLACEMENT varies by the renewable resource.”

Response from authors: this sentence now reads as follows: “At a systems-level, we observe negative correlations between renewable generation and both thermal generation and emissions, although the magnitude of these relationships varies by renewable resource.”

- p. 8: “In addition to the panel regression model that describes plant RESPONSE for the fleet [...] Overall, plants with a smaller nameplate capacity show a wider envelope of solar and wind coefficients IN RESPONSE TO increasing renewable generation.”

Response from authors: this sentence now reads as follows: “In addition to the panel regression model that characterizes plant-level relationships for the fleet, ...”

Overall, plants with a smaller nameplate capacity show a wider range of solar and wind coefficients in relation to renewable generation.

- p. 8: “In summary, we show that the first order EFFECT of increasing generation from renewables IS lowered emissions and generation from thermal power plants.”

Response from authors: this sentence now reads as follows: “In summary, we demonstrate that higher renewable generation corresponds to lower thermal plant generation and emissions.”

2) On p. 4, the authors write: “The model accounts for location and time fixed effects which allows us to interpret the coefficients independently from the plant fuel type.” I suggest replacing “location” with “plant” and ensuring consistent use of the term “plant fixed-effects” throughout the manuscript. The term “location” leaves unnecessary room for interpretation and could refer to coarser fixed effects than plant fixed-effects.

Response from authors: the term “location” has been replaced with plant wherever applicable, except in instances where we introduce the model as an “entity” and time fixed-effects model, which is the appropriate econometric representation.

3) The description in Table 2 includes the time fixed-effects but not the plant fixed-effects. But, as far as I understand, the regression results shown in Table 2 include plant fixed-effects. So I would suggest including them in the description as well.

Response from authors: the following statement has been added to the caption of Table 2: “Plant fixed-effects control for individual entity-based heterogeneity.”

4) In response to one of my initial comments, the authors provided an explanation of why they used their own measure of wind intermittency instead of simply the variance of wind generation in the analysis. It seems helpful to readers to also include a brief explanation of this when they introduce their measure in equation (2) in Methods.

Response from authors: the following statement has been added in the Methods section -
“The intermittency parameter \overline{W}_t captures the cumulative hourly fluctuations in wind generation throughout the day, rather than using a simpler measure like daily variance. This specification focuses on short-term variations between consecutive hours that may require rapid adjustments in thermal generation to maintain grid stability. While variance would measure overall daily deviations from mean wind generation, our intermittency measure specifically quantifies the magnitude of hour-to-hour changes that operators must balance. This is particularly relevant as frequent short-term fluctuations in wind generation may necessitate different operational responses from thermal plants compared to more gradual changes over the course of a day.”

Reviewer #4 (Remarks to the Author):

Response from authors: Thank you for your comments and feedback.

Remarks on formatting

The title has been changed to “Assessing the real implications of CO2 as generation from renewables increases” to avoid the usage of punctuation.

Parts a and b of Table 2 have been merged into one with the appropriate headers.

The lettering in the figures have been changed to reflect the size used elsewhere.

The Supplementary Information document has been appropriately reformatted, with all instances of headings numbers changed. Where referred to in the main text, Supplementary Note and Supplementary Figure have been used as opposed to SI.